# SAMPLE EFFICIENT FORCED DYNAMICS RECOVERY

## ABSTRACT

Recovering governing equations of dynamical systems from limited samples is critical for deploying autonomous systems under real-world resource constraints. Classical sparse regression methods (e.g., SINDy-MPC) and physics-informed neural networks achieve good fits when oversampled, but their accuracy degrades sharply when data is available only at the Nyquist rate. We provide an information-theoretic analysis showing that reconstruction error fundamentally decomposes into a data-fit component (linear in sampling frequency) and a model-estimation component (nonlinear in frequency), bounded by the Cramér–Rao lower bound. This motivates MRIDHA, a model recovery framework that constrains the equation search space to physically consistent structures by embedding continuous-time latent variable nodes that enforce stability and time-constant properties. Across nine simulated and three real-world benchmarks—including automated insulin delivery, EEG reconstruction, and a 16D quadcopter—MRIDHA significantly outperforms SINDy-MPC and PINN-SR at Nyquist-rate sampling, demonstrating improved sample efficiency, robustness to input uncertainty, and scalability. Our results establish both new theoretical limits and a practical method for sample-efficient recovery of forced dynamics.

## 1 INTRODUCTION

By definition, an autonomous system couples a device with a physical process through forcing inputs calibrated by sensor data. Assurance of autonomous system safety and reliability, often requires the development of a **replay simulator** that can reconstruct the behavior of the autonomous system for novel inputs and scenarios. The process of Model Recovery (MR) extracts an estimation of the underlying dynamical equations $(f, \theta)$ from multivariate trajectories $X(t)$ under forcing inputs $U$ obtained from real-world deployments Kaiser et al. (2018) that can be used by downstream replay simulators for comprehensive safety analysis. The goal of MR is to minimize **replay reconstruction error**, i.e., the discrepancy between true behavior and replayed behavior using the recovered equations for a variety of novel forcing inputs. This error arises from two sources: (a) **data fit error**, the mismatch between training scenario ground-truth samples and replay outputs, and (b) **model estimation error**, the failure to identify the correct functional structure $f$ and coefficients $\theta$.

Data collected from real-world systems is inherently constrained by storage, energy, privacy, and sampling limitations. Consequently, sample efficiency Gao et al. (2022)—the amount of data required for an AI module to learn and perform a task—is a critical consideration. In the context of MR, sample efficiency has two key dimensions: (a) *data length*, i.e., the duration of time-series data required from the system, and (b) *sampling frequency*, i.e., the number of measurements per unit time. Prior work has primarily addressed the impact of data length on reconstruction error in state-of-the-art (SOTA) MR methods through novel approaches Chen et al. (2021). In contrast, this paper focuses on the role of sampling frequency and introduces a new MR technique for forced dynamics that achieves improved reconstruction accuracy under low sampling regimes.

MR is a special case of the band-limited signal reconstruction (BSR) problem from uniformly sampled data Shannon (1949). The Nyquist–Shannon sampling theorem Shannon (1949) establishes that a signal can be reconstructed without information loss if it is sampled at or above the Nyquist rate, defined as twice the maximum frequency present in its spectrum. Consequently, any MR technique must operate in a manner consistent with this sampling requirement Vaidyanathan (2001).

A key limitation of state-of-the-art MR methods is their reliance on unrealistically high sampling rates. For example, sparse identification of nonlinear dynamics for model predictive control (SINDy-MPC) Kaiser et al. (2018) assumes sampling frequencies orders of magnitude above the Nyquist rate, while physics-informed neural networks with sparse regression (PINN-SR) Chen et al. (2021) mitigate data-length issues but still require very dense sampling. Figure 1 illustrates this effect: for a 1D sinusoidal system with maximum frequency 4Hz ($x(t) = \sin(\omega t), , \omega = 2\pi \times 4$), the reconstruction error of SINDy-MPC and PINN-SR grows sharply as the sampling frequency decreases from 512Hz

to the Nyquist rate of 8Hz and below. This demonstrates that existing methods fail to operate near the information-theoretic limits of sampling, motivating the need for improved MR techniques.

**Contributions:** Our contributions are:

**Contribution 1: Theoretical foundations of frequency-dependent reconstruction error.** We formalize MR as an unbiased estimator extraction problem Lin et al. (2005), where reconstruction error is bounded by the Cram'er–Rao Lower Bound (CRLB) Lin et al. (2005). Our information-theoretic analysis shows that the model estimation error grows nonlinearly as sampling frequency decreases, while the data fit error decreases linearly. Existing methods such as SINDy-MPC Kaiser et al. (2018) and PINN+SR Chen et al. (2021) (Figure 1, detailed in Figure 2) employ a two-step process: (a) unconstrained equation search—via least squares in SINDy-MPC or feedforward networks followed by SINDy in PINN+SR—and (b) post-hoc sparsity constraints that retain only dominant nonlinear terms. While sparsity captures some physical structure, the unconstrained search often yields dynamics that minimize data-fit error but violate ground-truth stability or physical laws, leading to high reconstruction error under novel forcing inputs.

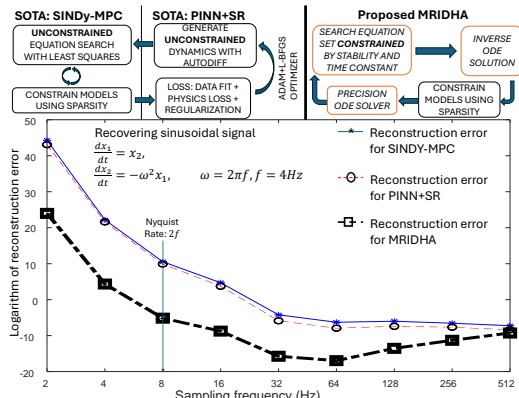

Figure 1: Effect of sampling frequency on reconstruction error. If sampling frequency decreases reconstruction error increases.

**Contribution 2: Novel MR technique for low sampling rates.** We propose MRIDHA (Model Recovery Infused with "DHArma"), which reduces reconstruction error by constraining the equation search space to models that respect physical properties such as stability and time constants. In MRIDHA (Figure 1, detailed in Figure 2), candidate dynamics are represented using a network of continuous-time latent variable (CTLV) nodes instantiated in a liquid time constant neural network (LTC-NN). We prove that the LTC-NN forward pass provides bilinear approximations of control-affine nonlinear dynamics. A subsequent dense layer maps these outputs to an over-determined system of algebraic equations, effectively solving the inverse ordinary differential equation (ODE) problem to estimate contributions of nonlinear library components. A sparsity threshold then prunes irrelevant outputs. The entire network is optimized using an *ODE loss*, defined as the mean square error between trajectories generated by an ODE solver and the ground truth measurements.

**Case study contribution:** We instantiate three architectures in MRIDHA using LTC-NN (MRIDHA-L), NODE (MRIDHA-N) and CT-RNN (MRIDHA-C) and compare with SOTA nonlinear MR techniques, SINDy-MPC Kaiser et al. (2018), PINN-SR Chen et al. (2021) among others. For benchmark case studies, we use three simulation and one real-world examples available in Kaiser et al. (2018). We introduce two more simulation and two more real world medical benchmarks with automated insulin delivery (AID) system and electro-encephalogram (EEG) brain signal reconstruction using nonlinear oscillator models Ghorbanian et al. (2015). For AID we create a simulation benchmark using the Food and Drug Administration (FDA) approved UVA/PADOVA Type 1 Diabetes (T1D) simulator Visentin et al. (2018). We also test our techniques on real-world clinical study using the publicly available LOIS-P dataset for pregnant women with T1D O'Malley et al. (2021). For EEG case study, we test all techniques on a simulated EEG dataset and real-world CHB-MIT Scalp EEG dataset Guttag (2010). To test scalability of MR techniques we have introduced a 16 D quadcopter example. In summary we evaluate on **six** simulation and **three** real-world case studies. In real world case studies, we tackle two important artifacts: a) **sensor noise**, where the data $X$ is corrupted with magnitude noise measured using the signal to noise ratio (SNR), and b) **sensor timing uncertainty**, where due to issues such as manual reporting error or time synchronization, the timing of inputs have errors.

## 2 PRELIMINARIES AND PROBLEM STATEMENT

We assume control affine system with $n$ dimensional state space $X = \{x_1 \ldots x_n\} \in \mathcal{R}^n$ given by:

$$\frac{dX}{dt} = K(X, \Theta, U) = f(X, \Theta) + g(X, \Theta)(U), \tag{1}$$

where $f(X, \Theta) : \mathcal{R}^n \times \mathcal{R}^p \to \mathcal{R}^n$ is a model of the natural unperturbed dynamics of the physical system that is perturbed by external inputs $U \in \mathcal{R}^n$. $g(X, \Theta) : \mathcal{R}^n \times \mathcal{R}^p \to \mathcal{R}^n \times \mathcal{R}^n$, expresses the effect of the input perturbation to the plant dynamics. $\Theta$ is a set of coefficients for the model of the autonomous system operation and $K(X, \Theta, U)$ is the overall dynamics. We use the control

Table 1: Related works. Sampling High is > Nyquist rate, Low is = Nyquist rate. Bold = baselines.

| Approach | Sampling | Inputs | Comments | Rationale for baseline |
|---|---|---|---|---|
| Ho Kalman, Eigen system Oymak & Ozay (2021) | Low | Yes | Linear system | Does not maintain model structure |
| Genetic Algorithm Schmidt & Lipson (2009) | High | No | Low dimension | Does not scale |
| SINDy Quade et al. (2018) | High | No | Unperturbed nonlinear dynamics | Cannot handle inputs |
| **SINDy-MPC** Kaiser et al. (2018) | **High** | **Yes** | **High error at low frequencies** | **Widely used** |
| E-SINDy Fasel et al. (2022) | Low | No | Only data length addressed | SINDy-MPC is E-SINDy+inputs |
| **W-SINDY** Messenger & Bortz (2021) | **High** | **No** | **Unperturbed nonlinear dynamics** | **Focuses on noise reduction** |
| **SVISE** Course & Nair (2023) | **Low** | **No** | **Unperturbed nonlinear dynamics** | **Focuses on low frequencies** |
| NODE metriplectic structure Lee et al. (2021) | High | No | Maintains metriplectic structure | Cannot extract model structure |
| **PINN Sparse Regression** Chen et al. (2021) | **High** | **No** | **Extracts model from initial value problem solutions.** | **Updated for forcing inputs** |
| **MRIDHA** | **Low** | **Yes** | **Black box ODE solver in loss** | **Proposed in this paper** |

affine assumption for the ease of explanation of the proofs in the manuscript. However, the proofs of the theorems also stand with $K(.,.,.)$. Henceforth we only focus on sub-problem of recovering $f(.)$. Examples of $K(.,.,.)$ are provided in Section E in appendix

**Problem Statement:** Given a time series of $X(t), U(t)$ sampled at frequency $f_s$, in this paper, we solve the MR problem to derive $\Theta_{est}$ such that the estimated signal $X_{est}$ has a reconstruction error $||X_{est} - X||^2 < \psi, \psi > 0$ on the training data.

**Cramer Rao Lower Bound:** In estimation theory, given $N$ samples of data $X(t_i)$, $i \in \{1 \dots N\}$ sampled at uniform rate / frequency $f_s$ at times $t_1 \dots t_N$, $t_i - t_{i-1} = 1/f_s \forall i \in \{1 \dots N\}$, the task is to derive the parameters $\Theta$ that defines the function $f(X, \Theta)$ and $g(X, \Theta)$. Cramer Rao bound provides the fundamental limits of error for estimation methods Lin et al. (2005). It states that the variance of the estimated model coefficients $var(\hat{\Theta})$ from an unbiased estimator satisfies the following inequality -

$$var(\hat{\Theta}) \geq \frac{1}{I(\Theta)}, \text{ such that } I(\Theta) = -\frac{1}{\sigma^2} \sum_{i=1}^{N} \max_{j \in \{1 \dots p\}} E\left(\frac{\partial^2 log(F(X(t_i), \theta))}{\partial \theta_j^2}\right), \quad (2)$$

where $I(\Theta)$ is the Fisher information, and $E(.)$ denotes expected value, and $F(X(t_i), \theta) = \int_0^{t_i} f(X(t_i), \theta) dt$. Fisher information is a fundamental property of the function $f$ or $g$ which is obtained by computing the Jacobian matrix with respect to $\Theta$. Eqn. 2 is a loose lower bound estimation of CRLB for unbiased estimator for Gaussian noise as proven in Handel (2000).

## 3 INFORMATION THEORETIC ANALYSIS OF MR

We perform an information theoretic analysis of CRLB, to explore its dependence on sampling frequency, and discuss key theoretical properties of MR reconstruction error.

### 3.1 CRLB AS A FUNCTION OF SAMPLING FREQUENCY

Integrating $f(X, \Theta)$, Fisher information is given by Eqn. 3 (derivation in appendix Section A).

$$I(\Theta) \leq \frac{1}{\sigma^2} \sum_{i=1}^{N} \sum_{k=1}^{Tf_s} \frac{1}{f_s} E\left(\frac{f''_{max}}{F(X(t_i), \theta)}\right), \text{ where } f''_{max} = \max_{\theta_j \in \Theta} \frac{\partial^2 f_{int}}{\partial \theta_j^2} = \max_{\theta_j \in \Theta} \max_i \frac{\partial^2 (\int_0^1 f(X(v/f_s + t_i), \Theta) dv)}{\partial \theta_j^2}, \quad (3)$$

where $T$ is the time horizon of data. Eqn. 3 shows that fisher information in each sample at time $t_i$ decreases with increasing frequency. However, as number of samples increase, the overall summation increases. Hence at low frequencies, increasing sampling frequency may increase the fisher information, however beyond Nyquist rate, the quantity $\frac{1}{f_s} E\left(\frac{f''_{max}}{F(X(t_i), \theta)}\right)$ becomes negligible and has no impact on $I(\Theta)$. This shows that the CRLB which is inverse of $I(\Theta)$ initially decreases with sampling frequency but asymptotically converges to a settling point $\lim_{f_s \to \infty} \frac{1}{f_s} E\left(\frac{f''_{max}}{F(X(t_i), \theta)}\right)$.

### 3.2 EFFECT OF FREQUENCY ON MODEL RECOVERY ERROR

*Remark* 1. The lower bound of the reconstruction error of MR methods have two components: a) data fit error, that is directly proportional to sampling frequency, and b) model estimation error, that increases with decreasing frequency following a non-linear relation.

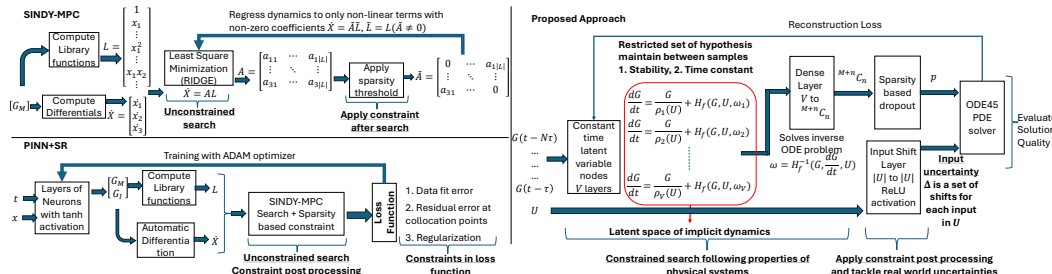

Figure 2: Learning architectures of SINDy-MPC, PINN+SR and MRIDHA that maintains time constant and stability properties of real world systems.

**Supporting Argument:** (full derivation in Remark 1 in appendix Section B) Assume that the maximum tolerable root mean square error on training data is $\psi$. Then the maximum reconstruction error at any sample at time $t$ within the horizon $T$, $g_e^{max}$ is given by:

$$g_e^{max} \leq \underbrace{f_s \psi}_{\text{data fit error}} + \underbrace{\sqrt{\sum_{i=1}^{N} \sup_{\Theta_{est}} f_{int}(X(t), \Theta_{est}) - \inf_{\Theta_{est}} f_{int}(X(t), \Theta_{est})/N}}_{\text{model divergence}}, \tag{4}$$

for $t \in [\frac{i}{f_s}, \frac{i+1}{f_s}]$. Eqn. 4 is obtained using residual analysis Ricchiuto & Abgrall (2010). $f_{int}$ is defined in Eqn. 3 as the integration of the function $f$. Eqn. 4 describes a trade-off between two objectives: a) accurate data fitting, and b) accurate model estimation. At higher sampling frequencies, the model estimation term is insignificant. If any MR technique solely focus on fitting the data, then it is highly probable that it will also find the right underlying dynamics. On the other hand, at low frequencies the model estimation term takes precedence over data fitting error. This indicates that at low frequencies the MR technique needs to focus on accurate model coefficient extraction, which guarantees good accuracy in data fitting.

## 4 LIMITATIONS OF SOTA MR SOLUTIONS

Table 1 summarizes recent MR methods. In the linear domain, system identification approaches such as Ho Kalman or the Eigen system realization algorithm (ERA) Oymak & Ozay (2021) fit linear models but cannot capture nonlinear dynamics or preserve sparsity. Early nonlinear recovery used symbolic regression and genetic programming Schmidt & Lipson (2009), but these methods failed to scale to higher dimensions.

**SINDy-MPC.** Sparse identification of nonlinear dynamics (SINDy) marked a breakthrough in MR and was later extended to handle control inputs in SINDy-MPC Kaiser et al. (2018). However, it performs poorly at low sampling frequencies. With $N$ samples at frequency $f_r = \frac{1}{\tau}$, the recovery reduces to solving: $dX_N/dt = \zeta \Theta_W$, where $X_N = \{X(\tau) \dots X(N\tau)\}^T$, $W = \binom{M+n}{n}$, and $\zeta$ is a matrix with $\zeta(X(k\tau), i)$ being the $i^{th}$ bilinear expansion term of $f(X)$ at $t = k\tau$. Since most $\theta_j \approx 0$ of the coefficient vector $\Theta_W$ for $j \in 1, \dots, W$, only $p$ coefficients remain significant ($\Theta = [\theta_1 \dots \theta_p]$). With $N \gg p$, the linear equation is over-determined, and $\Theta_{est}$ is obtained via least squares minimizing $e_T = ||X_{est} - X||^2$. SINDy-MPC solves this unconstrained problem using sequential threshold ridge regression (STRidge) Quade et al. (2018), which iteratively prunes nonlinear terms through hard thresholding of coefficients (Figure 2 left panel).

**PINN+SR:** As shown in Figure 2, PINN+SR Chen et al. (2021) approximates multivariate data $X$ with a feedforward network, computes $\frac{dX}{dt}$ via automatic differentiation, and evaluates a library of nonlinear functions. These functions are passed to SINDy, where STRidge produces a sparse coefficient matrix. The resulting ODE solution is refined by minimizing a combined loss of data-fit, physics residual, and regularization, optimized end-to-end with backpropagation.

**Other methods:** Neural ODE (NODE)-based approaches maintain metriplectic structures Lee et al. (2021), i.e., algebraic properties induced by physical laws such as energy conservation and thermodynamics Lee et al. (2021), but typically assume unrealistically high sampling rates.

**Limitations of SOTA MR:** As illustrated in Figure 2, SINDy-MPC applies sparsity only after unconstrained least-squares search via RIDGE regression, while PINN+SR searches for $G : X = G(t, \Theta)$ without enforcing structural validity and then applies SINDy-MPC constraints. Both approaches thus explore large unconstrained equation sets, often violating stability or time-constant properties, and may yield models that minimize data-fit error but remain physically infeasible.

## 5 MRIDHA MR TECHNIQUE

**Intuition:** According to Eqn. 4, to effectively reduce reconstruction error, MR techniques should explicitly incorporate model divergence error in the loss function. One way to reduce model divergence is to explicitly introduce $||\Theta - \Theta_{est}||^2$ in the loss function. While this is possible in simulation, the is not a practical solution since ground truth $\Theta$ is unknown. In MRIDHA, we take a *fundamentally different* approach to MR (Fig. 2 right panel), where we: **a)** first search for a function $H : X, U, \Theta_{est} \rightarrow dX/dt$ that models the original dynamics $dX/dt = K(X, \Theta, U)$ using a network of continuous time latent variable (CTLV) nodes, **b)** then solve the inverse ODE problem to obtain $\Theta_{est} = K^{-1}(dX/dt, X, U)$ using a standard dense layer, **c)** compute solution fitness *ODELoss*, by solving the ODE defined by $\Theta$ and evaluating mean square error between the solved ODE and the real world data, and **d)** using $ODELoss$ to train the entire network.

This approach (Fig. 2) introduces constraints at two levels: (a) **soft constraints**, restricting candidate functions $H$ to dynamics that satisfy stability and time-constant properties, and (b) **hard constraints**, enforcing sparsity on the inverse ODE solution $\Theta_{est}$. Soft constraints are realized through CTLV-based architectures—liquid time constant neural networks (LTC-NN), continuous-time recurrent neural networks (CT-RNN), and neural ODEs (NODE)—yielding MRIDHA-L, MRIDHA-C, and MRIDHA-N, respectively. NODE (MRIDHA-N) permits unconstrained integrable dynamics, CT-RNN (MRIDHA-C) restricts to stable dynamics with invariant time constant, and LTC-NN (MRIDHA-L) enforces stability with input-dependent time constants. Measurements of $X$ convert implicit dynamics into an overdetermined linear system for $\Theta$. a consistent solution is searcher via a dense layer. Hard constraints on the solution are applied by setting $\theta_i = 0$ when below a threshold. Training combines CTLV and dense layers with an *ODE loss*, defined as the mean square error between $X_{est} = \textbf{SOLVE}(X(0), \Theta, U)$ and the ground-truth $X$.

### 5.1 IMPLEMENTATION

MRIDHA is implemented by extending the base code available in Hasani (2024). For each example, we extract the training data consisting of temporal signals $X$, and $U$. $X$ is sampled at least at the Nyquist rate for the application, and $U$ has the same sampling rate as $X$. The resulting training data is then divided into batches of size $S_B$. This forms a 3 D tensor of size $S_B \times |X| + m \times k$.

Each batch is passed through the $\xi$ network with $V$ nodes, resulting in $V$ hidden states. A dense layer is then employed to transform this $V$ hidden states into $p = |\Theta|$ model coefficient estimates and $q$ input shift values. The dense layer is a multi-layer perceptron with ReLU activation function for the model coefficient estimate nodes and sigmoid activation function for input shift values. The input shift values are used to shift the external input vector. The shifted inputs, the model coefficient estimates, and the initial value $X(0)$ is passed through an ODE solver, that solves the control affine model in Eqn. 1 with the coefficients $\Theta_{est}$, initial conditions $X(0)$ and inputs $U$. The ODE45 solver is used, which gives $X_{est}$. In the backpropagation phase the network loss is appended with ODE loss, the mean square error between the original trace $X$ and estimated trace $X_{est}$.

### 5.2 THEORETICAL JUSTIFICATION FOR MRIDHA

We show that the CTLV layer can model any stable control affine dynamics with a time constant and the dense layer can accurately solve the inverse ODE problem.

**Theorem 1.** *For every hidden output of the LTC-NN layer of MRIDHA there exists a equivalent bilinear approximation of the control affine dynamics in Eqn. 1.*

**Proof Intuition:** The forward pass of LTC-NN network can be decomposed into an input dependent time constant component mimicking $g(., .)$ and a input independent dynamical component mimicking $f(., .)$. Full Proof in appendix Section B.

**Theorem 2.** *There exists a weight matrix $W_d \in R^{V \times \binom{M+n}{n}}$ of the dense layer such that it implements a solution to the inverse ODE problem of the original dynamics in Eqn. 1.*

**Proof intuition:** This result follows directly from the application of Universal Approximation Theorem where we can extract a weight matrix that expresses $\Theta$ as a function of $X$. Proof in appendix Section B.

**Why MRIDHA works?** Theorem 1 shows that at any stage of the learning MRIDHA only searches through dynamics that are stable and have an input dependent time constant. Theorem 2 shows that the dense layer is equivalent to solving the inverse ODE problem. These two theorems imply at each backpropagation step, MRIDHA attempts to simultaneously reduce both model estimation and data fit errors. This enables MRIDHA to reach closer to the CRLB.

## 6 EVALUATION

We evaluate: a) performance variation with frequency, b) performance variation at Nyquist frequency with noise, c) performance on real data, and d) ablation studies for MRIDHA-L components.

### 6.1 BENCHMARK EXAMPLES

Table 2 lists all benchmarks from Kaiser et al. (2018) (dynamics in appendix Section E ).

**Automated Insulin Delivery:** The glucose insulin dynamics is given by Bergman (2021) as:

$$\dot{i}(t) = -ni(t) + p_4 u_1(t), \dot{i_s}(t) = -p_1 i_s(t) + p_2(i(t) - i_b), \dot{G}(t) = -i_s(t)G_b - p3(G(t)) + u2(t)/VoI, \quad (5)$$

The input $U(t)$ includes insulin level $u_1(t)$, from a model predictive controller, and glucose appearance $u_2$ from a meal. Users may err in reporting meal timing or carbohydrate intake Lamrani et al. (2021); we model this as uncertainty in time and magnitude. $X(t)$ consists of blood insulin $i$, interstitial insulin $i_s$, and glucose $G$, with patient-specific coefficients $p_1, p_2, i_b, p_3, p_4, n$, and $1/V_oI$.

Real World Data In the real world AID example we used the LOIS-P dataset O'Malley et al. (2021), which comprised 25 patients with pre-existing T1D and at least 17 weeks pregnant. On an average 24.7 weeks ($\pm$ 5.2) of Dexcom G6 CGM glucose at 5 mins interval and insulin and meal intake data.

Simulation data for AID: We considered 14 traces of glucose insulin dynamics with 200 samples each. Meal ingestion time was varied from [t = 15 mins to t = 400 min] with carbohydrate value randomly sampled from the range $[0g, 28g]$, and bolus insulin was sampled from the set $[0U, 40U]$. The traces were generated using the T1D simulator Visentin et al. (2018)

Table 2: Benchmarks and novel datasets. Examples below the dashed lines are novel. S - Simulation, R - Real World

| Example | n-D | Inputs | Nyquist rate Hz | Max rate Hz | p |
|---|---|---|---|---|---|
| R: Lotka Volterra | 2 | 1 | 2.5 | 10 | 4 |
| S: Chaotic Lorenz | 3 | 1 | 100 | 1000 | 4 |
| S: F8 Crusader tracking | 3 | 1 | 100 | 1000 | 20 |
| S: Pathogenics attack | 5 | 1 | 0.00028 | 0.00056 | 13 |
| R: Auto Insulin Delivery | 3 | 2 | 0.0028 | 0.0033 | 9 |
| S: Auto Insulin Delivery | 3 | 2 | 0.0028 | 10 | 9 |
| S: EEG epilepsy | 4 | 1 | 250 | 500 | 6 |
| R: EEG epilepsy | 4 | 1 | 250 | 250 | 6 |
| S: Quadcopter | 16 | 4 | 125 | 500 | 37 |

with two sampling frequencies: i) 10Hz, and ii) real world sampling of every 5 mins (Table 2).

**EEG example:** Brain waves modeled as Duffing—van der Pol oscillators Ghorbanian et al. (2015).

$$\ddot{x}_1 + k_1 x_1 = k_2 x_2 - b_1 x_3^2 - b_2(x_1 - x_2)^3 + \epsilon_1 \dot{x}_1(1 - x_1^2), \quad (6)$$

$$\ddot{x}_2 - k_2(x_1 - x_2) = b_2(x_1 - x_2)^3 + \epsilon_2 \dot{x}_2(1 - x_2^2) + \mu dW,$$

where $k_1$, $k_2$, $b_1$, $b_2$, $\epsilon_1$ and $\epsilon_2$ are patient specific parameters, $x_2$ is the EEG signal, $x_1$ is an internal state variable, $\mu$ is the activation level, $dW$ is a random variable (Wiener process) Ghorbanian et al. (2015). Here the input $u = \mu dW$ has both temporal and magnitude uncertainty.

Simulation data: We developed a ODE45 based solver for the Eqn. 6 and generated data for two frequencies 250 Hz and 500 Hz.

Real World Data: The EEG example is evaluated with real world CHB-MIT Scalp EEG database Guttag (2010) with 684 EEG signals from 22 subjects at a sampling rate of 250 Hz.

**Quadcopter example:** To evaluate the scalability of MRIDHA on high dimensional systems we introduce the Quadcopter system that has 16 dimensions, four inputs corresponding to the four rotors, and 37 model coefficients.

Simulation data: We executed the quadcopter for 10 hover maneuvers at different heights with two sampling rates of 125 Hz and 250 Hz.

### 6.2 BASELINE TECHNIQUES

We compare with two sets of baselines: a) forced system baselines: **SINDy-MPC** Kaiser et al. (2018) and **PINN+SR** Chen et al. (2021) and b) unperturbed system low frequency high noise baselines: **W-SINDY** Messenger & Bortz (2021) and **SVISE** Course & Nair (2023). The unperturbed systems are evaluated on simulation benchmarks available in Course & Nair (2023) since nearly all real world systems have forcing inputs and unperturbed real world autonomous systems are rare.

**Ablation study:** Apart from our original model MRIDHA-L that uses LTC-NN nodes we test two more architectures, a) **MRIDHA-C,** this baseline uses CT-RNN layer instead of LTC-NN and tests the case if input dependent time constant is replaced by an invariant time constant, and b) **MRIDHA-N**, this uses NODE instead of LTC-NN, and tests case where time constant is removed.

### 6.3 EVALUATION EXPERIMENTS AND METRICS OF SUCCESS

For each evaluation experiment in simulation datasets, we use three metrics: **Mean absolute relative difference between ground truth and estimated coefficients as percentage of ground truth coefficients** ($MARD$),

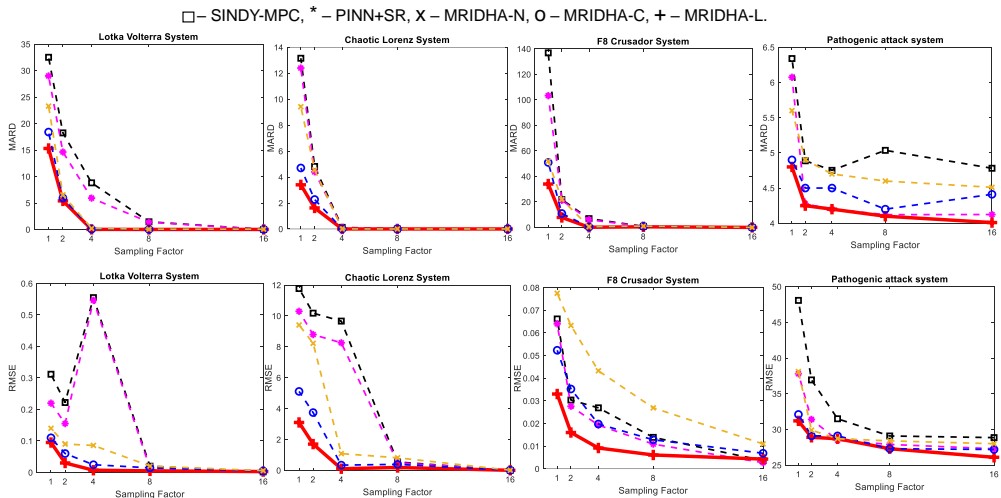

Figure 3: Performance comparison of MRIDHA with SINDy-MPC and PINN-SR by varying sampling frequency. The sampling frequency is equal to sample factor times Nyquist rate (Table 2).

hamming distance ($HD$), which matches the non-zero elements in ground truth $\Theta$ and estimated $\Theta_{est}$ and **Root mean square error in ground truth signal and estimated signal** ($RMSE$). Given the estimated model coefficients $\Theta_{est}$ and measured variables $Y_{est}$ we computed them using standard methods described in Eqn. 47 in appendix. For real wold, we can only use RMSE since $\Theta$ is not available.

Table 3: Hamming distance for benchmarks. Each entry has five numbers corresponding to sampling rates $f_s, 2f_s, \ldots 16f_s$ as in Fig 3.

| Method | Lorenz | Lotka | Pathogenic | F8 |
|---|---|---|---|---|
| SINDY-MPC | 4,4,3,2,0 | 3,4,3,1,0 | 9,13,6,3,0 | 12,9,5,3,0 |
| PINN+SR | 4,3,2,0,0 | 3,2,2,1,0 | 7,5,3,0,0 | 12,10,5,2,1 |
| MRIDHA-L | 2,2,2,0,0 | 3,1,0,0,0 | 7,5,2,1,0 | 9,8,5,2,0 |

**Experiments:** To address our evaluation goals we perform: *Experiment 1: To determine the effect of sampling rate on MR performance.* We consider the benchmark examples in Kaiser et al. (2018) without any sensor noise or timing error. We vary the sampling rate from the Nyquist rate (Table 2) and keep doubling it reaching 16 times the Nyquist rate. We analyze $MARD$ and $RMSE$.

Table 4: Sample efficiency for methods to keep MARD at $< 5\%$.

| Approach | Lotka Volterra | Chaotic Lorenz | F8 Crusador | Pathogenic |
|---|---|---|---|---|
| SINDyMPC | 20Hz | 200Hz | 400Hz | 0.00112Hz |
| PINN+SR | 20Hz | 200Hz | 400Hz | 0.00112Hz |
| MRIDHAC | 10Hz | 100Hz | 200Hz | 0.00028Hz |
| MRIDHAN | 10Hz | 200Hz | 400Hz | 0.00056Hz |
| MRIDHAL | 5Hz | 100Hz | 200 Hz | 0.00028Hz |

*Experiment 2: To determine the effect of magnitude noise at low frequency.* We keep the sampling frequency at Nyquist rate and introduce various additive Gaussian noise by varying SNR from 20dB down to 5 dB in steps of 5 dB and report $MARD$ and $RMSE$.

Table 5: Results for AID simulation example.

| Approach | $f_s = 10Hz$ | | | $f_s = 0.0033Hz$ | | | without input shifts | | |
|---|---|---|---|---|---|---|---|---|---|
| | $RMSE$ | $MARD$ | HD | $RMSE$ | $MARD$ | HD | $RMSE$ | $MARD$ | HD |
| SINDyMPC | 0.004(0.003) | 3.4(1.2) | 1 | 14.5(2.6) | 24.4(14) | 3 | 101(14) | 223(73) | 6 |
| PINN+SR | 0.003(0.002) | 2.7(1.0) | 0 | 4.5(1.7) | 13.4(8) | 3 | 83(17) | 189(67) | 7 |
| MRIDHAC | 0.007(0.002) | 3.1(1.0) | 0 | 0.76(0.07) | 8(2) | 3 | 43(10) | 178(43) | 4 |
| MRIDHAN | 0.012(0.006) | 5.6(1.7) | 0 | 1.3(0.33) | 11(2) | 3 | 77(11) | 236(71) | 5 |
| MRIDHAL | 0.003(0.001) | 2.1(0.9) | 0 | 0.31(0.9) | 4.5(1.9) | 2 | 31(5) | 141(61) | 3 |

*Experiment 3: Evaluating MR performance under real world settings:* We compare all approaches for various degrees of input timing uncertainty in a simulation setting. Then we evaluate all techniques on real data. In these experiments, the SOTA baselines are at a disadvantage since they do not use input shifts to address timing uncertainty. Hence for a fair comparison we also show results for disabling input shifts. We do not know the ground truth model coefficients $\Theta$ and hance we compare the techniques using $RMSE$ only.

In each of the experimental settings we report performance of MRIDHA-C and MRIDHA-N for ablation studies to see effect of removing time constant and stability constraints.

## 6.4 TRAINING AND VALIDATION METHOD

**SINDy-MPC:** We utilize the same training method as used in SINDy-MPC (code reused from Kaiser (2024)). In the experiments, to simulate frequency change, we varied the `dt` variable in the "getTrainingData.m" to the values that correspond to each frequency setting, $dt = 1/f$ (Table 2). The Nyquist rate is obtained by computing the power spectral density of the solutions of the ODE. We then extract the frequency $f_{90}$ at which the cumulative power density reaches 90% of the maximum level. The Nyquist rate is two times $f_{90}$.

**MRIDHA / PINN+SR Architecture:** Batch training was utilized for each example. For each example, we took the same simulation data as SINDy-MPC and divided the traces into 48 instance of training and 16 instance of test each of at least $k = 200$ samples. These training instances were passed to the network architectures with a batch size $S_B = 32$. The $RMSE$ and $MARD$ are reported on the test data. Code and implementation details in appendix Section C.

## 6.5 RESULTS

**EXPERIMENT 1: Effect of sampling rate.** As seen from Fig. 3 and Table 3, at sampling rates nearly 16 times Nyquist rate, all techniques give similar $MARD$, $RMSE$, and $HD$.

All neural architectures outperform SINDy-MPC, with MRIDHA-L achieving the best results, surpassing PINN+SR. This is because SINDy-MPC enforces only sparsity, while MRIDHA adds stricter constraints on the equation search. As sampling rates decrease, performance degrades for all methods, but SINDy-MPC is most affected. The strongest constraint, the input-dependent time constant in MRIDHA-L, yields the best performance, followed by MRIDHA-C and then NODE. Except for the pathogenics attack, MRIDHA-L consistently outperforms all neural architectures and SINDy-MPC at Nyquist sampling. For the pathogenics attack example, an interest-

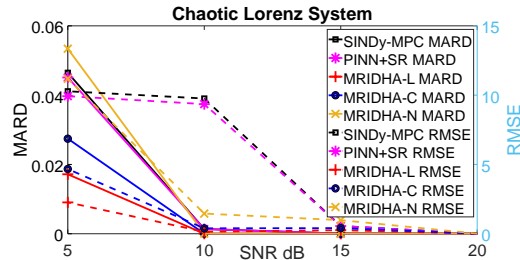

Figure 4: MRIDHA performance comparison for different noise levels for the Lorenz system.

ing occurrence is observed, where at Nyquist rate, SINDy-MPC has the best performance in both the metrics. However, at the next sampling frequency we see SINDy-MPC has a very high $RMSE$ for a slight change in $MARD$. All neural architectures differed from this trend and both $MARD$ and $RMSE$ improved. On closer look we found that SINDy-MPC found a totally different physical model of the plant with a high $HD$.

This is also seen in the LOTKA-Volterra and F8 Crusader example, where decreasing sampling frequency to Nyquist rate reduced $RMSE$ but increased $MARD$. Similarly it was observed that SINDy-

Table 6: $RMSE$ comparison for AID real world example with uncertain meal timing and sensor noise.

| Approach | no input shifts | with input shifts | Approach | $RMSE$ |
|---|---|---|---|---|
| MRIDHAN | 45.6 (23.2) | 8.7 (6.1) | SINDyMPC | 73.4 (27.1) |
| MRIDHAC | 32.3 (13) | 6.8 (4.4) | PINN+SR | 58.5 (17.9) |
| MRIDHAL | 26.1 (6.2) | 3.03 (0.4) | - | - |

MPC compensated for loss in $MARD$ performance by adding extra non-linear terms to reduce $RMSE$. From the results, we do not see such behavior for MRIDHA. One main factor is that the ODE loss guides the exploration of physically feasible dynamics. PINN+SR only mildly improved SINDy-MPC performance and followed similar trends to SINDy-MPC across all examples.

**Sample efficiency:** Table 4 shows the number of samples required by each method to achieve a MARD of $< 5\%$. We see that MRIDHA-L requires the least frequency among all techniques. We also see that PINN+SR is only slightly better than

Table 7: Performance for EEG simulation

| Approach | Sine input 500Hz | | | Sine input 250 Hz | | | Wiener input 500Hz | | |
|---|---|---|---|---|---|---|---|---|---|
| | $RMSE$ | $MARD$ | $HD$ | $RMSE$ | $MARD$ | $HD$ | $RMSE$ | $MARD$ | $HD$ |
| SINDYMPC | 0.1 | 0.21 | 2 | 23.2 | 46.1 | 7 | 144.1 | 101.3 | 14 |
| PINN+SR | 0.08 | 0.19 | 1 | 17.6 | 26.1 | 7 | 140.7 | 85.3 | 11 |
| MRIDHAN | 0.13 | 0.21 | 2 | 12.9 | 10.1 | 9 | 73.4 | 26.1 | 8 |
| MRIDHAC | 0.12 | 0.205 | 2 | 9.4 | 7.9 | 8 | 27.9 | 17.1 | 9 |
| MRIDHAL | 0.1 | 0.203 | 1 | 6.3 | 4.7 | 7 | 19.8 | 12.9 | 8 |

SINDY-MPC in terms of MARD and has no improvement on sample efficiency.

**EXPERIMENT 2: Effect of noise on MR accuracy.** Fig. 4 shows for the Lorenz system that as SNR decreases, all techniques perform

Table 8: $RMSE$ for EEG real-world example

| SINDY-MPC | PINN+SR | MRIDHAN | MRIDHAC | MRIDHAL |
|---|---|---|---|---|
| 1211.3(489.1) | 578.4(119.1) | 123.1(45.6) | 47.9(26.1) | 41.2(27.9) |

poorly with respect to both MARD and RMSE (All other benchmarks in Fig. 5 in appendix). MRIDHA-L approach provides a stable rise, while SINDy-MPC or PINN+SR show more variance.

**EXPERIMENT 3: Automated Insulin Delivery Example**

**Simulation Examples:** In Table 5, temporal uncertainty for meal inputs was not used for SINDy-MPC. All MRIDHA architectures had temporal uncertainty at meal inputs and also used input shifts. However, for the last column, the input shift from MRIDHA are removed and both SINDy-MPC and PINN+SR are also evaluated for uncertainty at meal input. All techniques perform well on simulation data at 10 Hz sampling rate. SINDy-MPC shows excellent $RMSE$ but poor $MARD$. MRIDHA performs similar to SINDy-MPC at such high sampling frequency. However, when the sampling rate is reduced to Nyquist rate, all methods have performance degradation, with SINDy-MPC suffering the most. MRIDHA-L still performs better than the baseline techniques. Without input shifts all techniques suffered significant reduction in performance.

**Real World Example Input Uncertainty:** Table 6 shows the performance of all approaches on real data. Without exploring temporal uncertainty of inputs, the best $RMSE$ for

Table 9: RMSE/MARD/HD for Quadcopter example

| SINDY-MPC | | PINN+SR | | MRIDHA | |
|---|---|---|---|---|---|
| 16F | F | 16F | F | 16F | F |
| 3.12/13.5/4 | 13.4/31.6/8 | 3.05/13.2/3 | 12.8/30.1/9 | 3.12/13.1/3 | 10.1/24.3/6 |

MRIDHA-L was 26.1. The SOTA CGM prediction mechanism for 30 mins ahead prediction has an RMSE of 11.1 Deng et al. (2021). With input shift enabled, we see significant improvement in $RMSE$ for MRIDHA architectures. MRIDHA-L has an $RMSE$ is 3.03 which is better than forecasting methods.

**EXPERIMENT 3: EEG reconstruction example**

**Simulation Example:** In simulation, we use a sinusoidal input as activation with two frequencies 250 Hz and 500 Hz and then the Wiener process in Ghorbanian et al. (2015) at 500 Hz. Table 7 shows that both SINDy-MPC and

Table 10: Performance comparison (RMSE/MARD/HD) at SNR = 5 dB for baselines without forcing inputs.

| System | SINDY | PINN+SR | wSINDY | MRIDHAL | SVISE |
|---|---|---|---|---|---|
| Vanderpol-16F | 1.1│2.3│2 | 0.9│1.7│0 | 0.8│1.4│0 | 0.7│1.3│0 | 0.4│12.3│8 |
| Vanderpol-F | 5.4│11│8 | 4.9│10.7│4 | 5.5│12.1│9 | 4.9│10.6│1 | 4.4│14│2 |
| Rossler-16F | 20.1│19.3│3 | 18.6│18.6│2 | 18.6│18.7│2 | 18.6│17.6│0 | 18.5│21.1│16 |
| Rossler-F | 23.1│24.2│18 | 23.0│23.9│14 | 27.6│27.4│15 | 24.5│23.6│10 | 25.2│36.1│13 |
| Oregonator-16F | 29.4│32.1│56 | 23.6│32.1│49 | 23.2│34.1│55 | 22.9│27.3│27 | 22.7│28.1│95 |
| Oregonator-F | 34.6│41.7│70 | 29.7│38.6│62 | 21.8│26.7│43 | 22.0│27.3│44 | 24.3│29.4│74 |

MRIDHA-L have comparable accuracy in extracting the model coefficients for high frequency data. However, at low frequencies measurements, we see that SINDy-MPC and PINN+SR recover entirely wrong model with high $MARD$, whereas MRIDHA-L has much lower $MARD$. Interestingly, if we use the Wiener process as input, then even at lower frequencies SINDy-MPC or PINN+SR recover a wrong models, which is not the case for MRIDHA-L.

**Real-World Example:** Table 8 shows that MRIDHA-L can replicate the EEG signal with much lower $RMSE$ than baselines for the CHB-MIT Scalp EEG dataset.

**Scalability to high dimensional system:** Table 9 shows that MRIDHA outperforms existing techniques for high dimensional systems such as the Quadcopter.

**Performance comparison for unperturbed systems:** We compared SINDY-MPC, PINN+SR and MRIDHA-L with SVISE which is specifically designed to handle low frequency data, and W-SINDY that has better performance than SINDY-MPC under high noise. Since both W-SINDY and SVISE methods recover models for unperturbed systems, we executed SINDY-MPC, PINN+SR and MRIDHA on new case studies listed in Messenger & Bortz (2021) from data collected at 5dB SNR for two frequencies at and 16 times Nyquist rate. Table 10 shows that MRIDHA has similat performances to SOTA MR techniques for unperturbed systems often outperforming the baselines.

**Reconstruction time:** For one epoch and batch size 1, MRIDHA's forward-pass complexity is $\mathcal{O}\big(Nn P_{\text{LTC}} + P_{\text{dense}}(\|\Theta\| + n)n + \text{ODESolve}\big)$, and the backward-pass complexity is $\mathcal{O}\big(P_{\text{LTC}} P_{\text{dense}} + P_{\text{dense}}(\|\Theta\| + n)\big)$, where $P_{\text{LTC}}$ and $P_{\text{dense}}$ are the numbers of LTC-NN and dense-layer nodes, respectively, and ODESolve denotes the time to solve one $n$-dimensional ODE. The STRIDGE algorithm complexity in SINDY-MPC is $\mathcal{O}\big(Q N n (\|\Theta\| + n)\big)$, where $Q$ is the number of sequential threshold executions. MRIDHA and SINDY-MPC scale similarly with respect to $n$, but MRIDHA is slower than SINDY-MPC due to the ODE solve per epoch, and PINN+SR is even slower. We measured reconstruction time (in seconds) on an NVIDIA RTX 6000 Ada GPU tower. SINDY-MPC is the most scalable, while MRIDHA-L achieves the best accuracy and is faster than PINN+SR. Full table of reconstruction times is provided in appendix (Table 11)

## 7 CONCLUSIONS

This paper observes an important relation of information theoretic lower bound of MR reconstruction error with frequency establishing two distinct components: a) data fit error that is dominant at high frequencies and b) model estimation error, that is dominant at low frequencies. We provided a fundamental sampling boundary, divergence boundary, which is useful in practical applications in presence of sampling and computational resource constraints. We have also provided a novel method for MR, MRIDHA, that improves reconstruction error at sampling frequencies lower than the divergence boundary by constraining the equation set search using prior information about stability and time constant of the system. MRIDHA outperforms SINDy-MPC and PINN+SR techniques in both simulation and real world case studies. We show that our proposed MRIDHA-L technique is robust against human reporting errors and provides a practical solution to a fundamental problem of generalized model recovery under sampling constraints. While majority of the examples fit the control affine definition, the F8 cruiser example is not control affine. Application of our technique in the F8 cruiser example shows generality beyond control affine systems.

## ETHICS STATEMENT

The results of the project are expected to have a profound impact onto health and wellbeing of our society by providing a pathway towards development of digital twins calibrated with real data. While this is a significant technological advancement, researchers should be aware of ethical issues related to data privacy while calibrating digital twins and potential impersonation attacks. Careful handling of model recovery techniques through institutional review boards is necessary to prevent such malicious purposes.

## REPRODUCIBILITY STATEMENT

The code is available at the anonymyzed link `https://anonymous.4open.science/r/LTC-NN-MR-E24C/`.

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

## A    PROOF OF CRAMER RAO LOWER BOUND

We start from Eqn 1 in the original paper for the unperturbed system and expand $X(t_i)$ as follows:

$$X(t_i) = X(t_{i-1}) + \int_{t_{i-1}}^{t_{i-1}+1/f_s} f(X(t), \Theta) dt \tag{7}$$

Using the variable replacement $p = f_s(t - t_{i-1})$, we get:

$$X(t_i) = X(t_{i-1}) + \frac{1}{f_s} \int_0^1 f(X(p/f_s + t_{i-1}), \Theta) dp \tag{8}$$

Starting from the $i^{th}$ sample if we move backwards in time we obtain the following set of equations:

$$X(t_i) = X(t_{i-1}) + \frac{1}{f_s} \int_0^1 f(X(p/f_s + t_{i-1}), \Theta) dp \tag{9}$$

$$X(t_{i-1}) = X(t_{i-2}) + \frac{1}{f_s} \int_0^1 f(X(p/f_s + t_{i-2}), \Theta) dp \tag{10}$$

$$\vdots \tag{11}$$

$$X(t_1) = X(t_0) + \frac{1}{f_s} \int_0^1 f(X(p/f_s + t_0), \Theta) dp \tag{12}$$

Adding all equations we get

$$X(t_i) = X(t_0) + \sum_{j=1}^{i-1} \frac{1}{f_s} \int_0^1 f(X(p/f_s + t_{i-j}), \Theta) dp \tag{13}$$

By performing double differentiation of Eqn. 13 we obtain Eqn. 14.

$$\frac{\partial^2 F(X(t_i), \Theta)}{\partial \theta_k^2} = \frac{1}{f_s} \sum_{j=1}^{i} \frac{\partial^2}{\partial \theta_k^2} \left( \int_0^1 f(X(v/f_s + t_{i-j}), \Theta) \, dv \right) \tag{14}$$

We consider Eqn. 2 in the main paper and compute the differentiation to get Eqn. 15

$$|I(\theta)| = \frac{1}{\sigma^2} \sum_{i=1}^{N} \max_{j \in \{1...p\}} E\left( \frac{1}{X(t_i)} \frac{\partial^2 (X(t_i))}{\partial \theta_j^2} \right) \tag{15}$$

We then substitute Eqn. 14 to Eqn. 15 to obtain Eqn. 16

$$
\begin{aligned}
|I(\theta)| &= \frac{1}{\sigma^2} \sum_{i=1}^{N} \max_{j \in \{1...p\}} E\left( \frac{1}{X(t_i)} \sum_{k=1}^{i} \frac{1}{f_s} \frac{\partial^2 \int_0^1 f(X(v/f_s + t_{i-k}), \Theta) \, dv}{\partial \theta_j^2} \right) \\
&\leq \frac{1}{\sigma^2} \sum_{i=1}^{N} \sum_{k=1}^{i} \frac{1}{f_s} E\left( \frac{1}{X(t_i)} \max_{j \in \{1...p\}} \frac{\partial^2 \int_0^1 f(X(v/f_s + t_{i-k}), \Theta) \, dv}{\partial \theta_j^2} \right) \\
&\leq \frac{1}{\sigma^2} \sum_{i=1}^{N} \sum_{k=1}^{i} \frac{1}{f_s} E\left( \frac{f''_{max}}{X(t_i)} \right)
\end{aligned}
\tag{16}
$$

# B SUPPORTING ARGUMENTS FOR REMARKS

*Remark* 1. The lower bound of the reconstruction error of MR methods have two components: a) data fit error, that is directly proportional to sampling frequency, and b) model estimation error, that increases with decreasing frequency following a non-linear relation.

**Supporting Argument:** Assume that the least square minimization problem is solved with a maximum error of $\psi : e_T \leq \psi$. This indicates that given two consecutive samples of $X$, $X(k\tau)$ and $X((k+1)\tau)$ we have the following inequality:

$$||X(k\tau) - X_{est}(k\tau)||^2 \leq \frac{\tau\psi}{T}, ||X((k+1)\tau) - X_{est}((k+1)\tau)||^2 \leq \frac{\tau\psi}{T}. \tag{17}$$

Within the sampling time interval, any recovered model with the guarantee of $\psi$ error can deviate from the original model. Given that we are only interested in sparse models, the deviation can be characterized as a deviation in model coefficients $\Theta$. Hence, in each time $t$ in between samples, the error in any unseen time $t$, in other words *reconstruction error* can be expressed as:

$$g_e(t) = || \int_{k\tau}^{t} [f(X(t), \Theta) - f(X(t), \Theta_{est})] dt ||^2, t \in [k\tau, (k+1)\tau] \tag{18}$$

subject to: $g_e(t) \leq \frac{\tau\psi}{T}$ for $t = k\tau$ and $t = (k+1)\tau$.

Considering the worst case difference in ground truth and estimated model coefficients we get an upper bound of this reconstruction error subject to the constraints in Eqn 17:

$$g_e^{max} \leq \underbrace{f_s \psi}_{\text{data fit error}} + \underbrace{\sqrt{\frac{\sum_{i=1}^{N} \sup_{\Theta_{est}} f_{int}(X(t), \Theta_{est}) - \inf_{\Theta_{est}} f_{int}(X(t), \Theta_{est})}{N}}}_{\text{model divergence}}, \tag{19}$$

for $t \in [\frac{i}{f_s}, \frac{i+1}{f_s}]$. Eqn. 19 is obtained using residual analysis Ricchiuto & Abgrall (2010).

**Theorem 1.** *For every hidden output of the LTC-NN layer of MRIDHA there exists a equivalent bilinear approximation of the control affine dynamics in Eqn. 1.*

**Proof:**

Algebraic manipulation of the forward pass of LTC-NN architecture gives the structure of Eqn. 20 which allows an input dependent time constant $\frac{\rho}{1+\rho f_{NN}(h(t), I(t), t, \omega)}$.

$$\frac{dh(t)}{dt} = -\frac{h(t)}{\frac{\rho}{1+\rho f_{NN}(h(t),I(t),t,\omega)}} + f_{NN}(h(t), I(t), t, \omega)(A). \tag{20}$$

The stability criteria for any autonomous system requires the control affine model to have a time constant term as shown in Eqn. 21

$$\frac{dX}{dt} = -X/\rho + f_{-\rho}(X) + g(X)U_T, \tag{21}$$

where $\rho$ is the time constant of the system and $f_{-\rho}(.)$ is the unperturbed dynamics obtained by removing the time constant component from $f(.)$.

Assuming that the autonomous system is a dynamic causal system, the bilinear approximation Friston et al. (2003) of the control affine system in Eqn. 21 results in Eqn. 22.

$$\frac{dX}{dt} \approx -X/\rho + f_{-\rho}(X) + BX + CU_T + \sum_j u_T^j D^j X + H, \tag{22}$$

where $B = \frac{\partial(g(X)U_T)}{\partial X}$, $C = \frac{\partial(g(X)U_T)}{\partial U_T}$, and $D^j = \frac{\partial^2(g(X)U_T)}{\partial X \partial u_T^j}$, $H$ is a constant. Rearranging Eqn. 22, we have the similar form as the LTC-NN forward pass in Eqn. 23.

$$\frac{dX}{dt} \approx -\frac{X}{\frac{\rho}{1+\rho(B+\sum_j u_T^j D^j)}} + (f_{-\rho}(X) + CU_T + H). \tag{23}$$

We observe that Eqn. 23 is the same form as Eqn. 20 if the input to the LTC-NN $I(t)$ is a concatenation of $Y$ and $U_T$. The hidden layers of the LTC-NN model an inflated set of implicit dynamics which may include the unmeasured system variables of the physics model.

**Theorem 2.** *The inflated set of implicit dynamics modeled by LTC-NN induces an over-determined set of equations in the coefficients of the bilinear approximation of control affine model.*

**Proof:**

The training process of LTC-NN fixes weights and instantiates the hidden layer outputs. The values of the unmeasured variables in $X$ is estimated by the hidden state in each training step utilizing the forward pass and learned LTC-NN weights $\omega$. Hence each forward pass provides an over-determined set of linear equations in the coefficients $B$, $C$, and $D^j$.

The original control affine model coefficients $\Theta$ are non-linear functions of the coefficients $B$, $C$, and $D^j$s, The dense layer is best suited for exploring a large set of possible non-linear combinations of $B$, $C$, and $D^j$ that express $\Theta$. An overdetermined system of equations is inconsistent and may be unsolvable. The dense layer guided by the ODE solver induced loss function (ODE Loss) learns a consistent set of linear equations in $B$, $C$, and $D^j$ and also learns their non-linear combination to determine $\Theta$.

## C  IMPLEMENTATION DETAILS

For the neural architectures, we used based our implementation on the codebase available at Hasani (2024). Here a generic framework for LTC-NN, CT-RNN, and NODE is implemented using tensorflow 2.7.0. We wrote a custom loss function (code available in appendix) that implements the ODE45 solution of the physical dynamics given a vector of model coefficients. We use the general training architecture presented in Hasani (2024) with an ADAM optimizer. The framework can be instantiated with LTC-NN, CT-RNN and NODE core architecture through an input parameter.

## D  ADDITIONAL RESULTS

## E  ACTUAL V.S. RECOVERED MODELS

We provide the actual model and the recovered model while sampling at the divergence boundary of all the examples. The reconstruction error threshold used is $10\%$. All coefficients are rounded up to three decimal points after most significant digit.

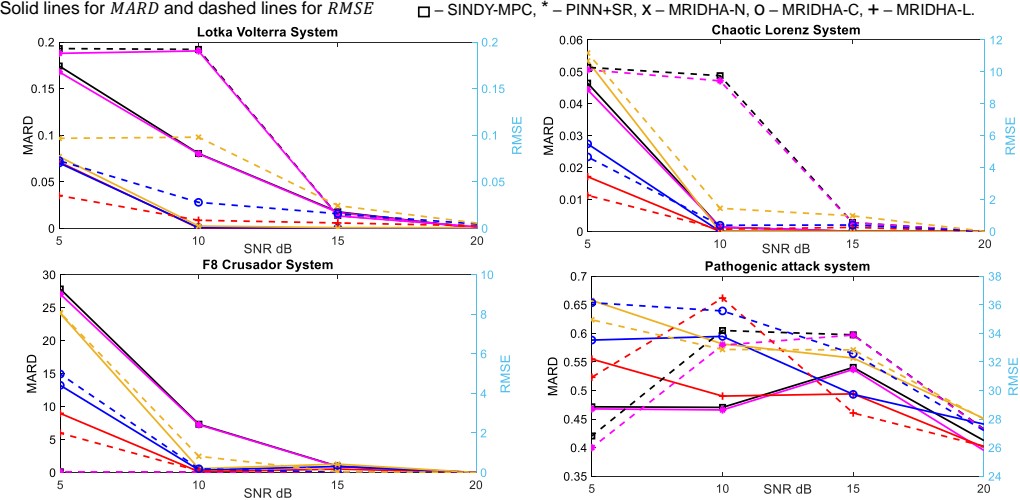

Figure 5: Comparison of SINDy-MPC with neural architecture for various noise levels.

## E.1  D. LOTKA VOLTERRA MODEL

It has two variables $x_1$ and $x_2$ given by the following equations:

$$\dot{x_1} = ax_1 - bx_1x_2 \tag{24}$$
$$\dot{x_2} = -cx_2 + dx_1x_2 + u \tag{25}$$

a = 0.5, b = 0.025, c = 0.5, and d = 0.005

**Recovered model:**

$$\dot{x_1} = 0.52x_1 - 0.026x_1x_2, \ \dot{x_2} = -0.501x_2 + 0.005x_1x_2 + 0.999u \tag{26}$$

## E.2  E. CHAOTIC LORENZ SYSTEM

The chaotic lorenz system is described in the following equations:

$$\dot{x_1} = \sigma(x_2 - x_1) + u \tag{27}$$
$$\dot{x_2} = x_1(\rho - x_3) - x_2 \tag{28}$$
$$\dot{x_3} = x_1x_2 - \beta x_3 \tag{29}$$

$\sigma = 10, \beta = 8/3, \rho = 28.$

**Recovered model:**

$$\dot{x_1} = 10.000(x_2 - x_1) + 0.999u, \ \dot{x_2} = 27.992x_1 - 1.002x_1x_3 - 0.998x_2, \ \dot{x_3} = 1.000x_1x_2 - 2.7x_3 \tag{30}$$

## E.3  F. F8 CRUSER SYSTEM

The F8 Cruser system is given by

$$\dot{x_1} = -0.877x_1 + x_3 - 0.088x_1x_3 \tag{31}$$
$$+0.47x_1^2 - 0.019x_2^2 - x_1^2x_3 \tag{32}$$
$$+3.846x_1^3 - 0.215u + 0.28x_1^2u \tag{33}$$
$$+0.47x_1u^2 + 0.63u^3 \tag{34}$$
$$\dot{x_2} = x_3 \tag{35}$$
$$\dot{x_3} = -4.208x_1 - 0.396x_3 \tag{36}$$
$$-0.47x_1^2 - 3.564x_1^3 \tag{37}$$
$$-20.967u + 6.265x_1^2u \tag{38}$$
$$+46x_1u^2 + 61.1u^3 \tag{39}$$

**Recovered model:**

$$\dot{x_1} = -0.872x_1 + 0.998x_3 - 0.088x_1x_3 + 0.476x_1{}^2 - 0.0186x_2{}^2 - 0.970x_1{}^2x_3 \tag{40}$$
$$+3.849x_1{}^3 - 0.22u + 0.265x_1{}^2u + 0.472x_1u^2 + 0.63u^3$$
$$\dot{x_2} = 1.000x_3$$
$$\dot{x_3} = -4.210x_1 - 0.399x_3 - 0.465x_1{}^2 - 3.565x_1{}^3 - 20.978u + 6.267x_1{}^2u + 45.711x_1u^2 + 62.002u^3$$

### E.4 G. PATHOGENIC ATTACK SYSTEM

The pathogenic attack system is given by the following equations:

$$\dot{x_1} = \lambda - dx_1 - \beta(1 - \eta u)x_1x_2 \tag{41}$$
$$\dot{x_2} = \beta(1 - \eta u)x_1x_2 - ax_2 - p_1x_4x_2 - p_2x_5x_2 \tag{42}$$
$$\dot{x_3} = c_2x_1x_2x_3 - c_2qx_2x_3 - b_2x_3 \tag{43}$$
$$\dot{x_4} = c_1x_2x_4 - b_1x_4 \tag{44}$$
$$\dot{x_5} = c_2qx_2x_3 - hx_5, \tag{45}$$

with $\lambda = 1$, $d = 0.1$, $\beta = 1$, $a = 0.2$, $p_1 = 1$, $p_2 = 1$, $c_1 = 0.03$, $c_2 = 0.06$, $b_1 = 0.1$, $b_2 = 0.01$, $q = 0.5$, $h = 0.1$, and $\eta = 0.9799$.

**Recovered model:**

$$\dot{x_1} = 0.939 - 0.1x_1 - 0.982x_1x_2 + 0.980ux_1x_2, \tag{46}$$
$$\dot{x_2} = 0.982x_1x_2 - 0.980ux_1x_2 - 0.18x_2 - 1.000x_4x_2 - 1.001x_5x_2$$
$$\dot{x_3} = 0.059x_1x_2x_3 - 0.03x_2x_3 - 0.009x_3$$
$$\dot{x_4} = 0.029x_2x_4 - 0.1x_4 \quad \dot{x_5} = 0.059x_2x_3 - 0.1x_5$$

## F EVALUATION METRICS

$$MARD = 100/p \times \sum_{j=1\dots p} \frac{abs(\Theta_{est}^j - \Theta^j)}{\Theta^j}, RMSE = \frac{1}{n} \sum_{l=1\dots n} \sqrt{\frac{1}{k} \times \sum_{j=1\dots k} (X_{est}^l(j) - X^l(j))^2}, \tag{47}$$

$$HD = \sum_{i=1}^{p} \left| \mathbf{1}_{\{\Theta^i \neq 0\}} - \mathbf{1}_{\{\Theta_{est}^i \neq 0\}} \right|, \text{ where } \mathbf{1}_{\{.\}} \text{ is indicator function} \tag{48}$$

## G RECONSTRUCTION TIMES

Reconstruction times are provided in Table 11.

## H CODE AVAILABILITY

The code is available at the anonymized link `https://anonymous.4open.science/r/LTC-NN-MR-E24C/`.

Table 11: MR reconstruction times (seconds).

| Method | Lorenz | Lotka | Pathogenic | AID | EEG | F8 | Quadcopter |
|---|---|---|---|---|---|---|---|
| SINDY-MPC | 52 | 91 | 311 | 218 | 512 | 783 | 1912 |
| PINN+SR | 512 | 721 | 5151 | 3795 | 9131 | 10511 | 25112 |
| MRIDHA-L | 132 | 214 | 4166 | 2411 | 4414 | 5133 | 12810 |

