# OpenReview forum: "Sample Efficient Forced Dynamics Recovery"
_ICLR.cc/2026/Conference — ICLR 2026 Conference Withdrawn Submission_

### Official Review · Reviewer_pn6u · 2025-10-29

**Soundness:** 2
**Presentation:** 1
**Contribution:** 2
**Rating:** 4
**Confidence:** 3

**Summary:**

This paper proposes MRIDHA, a model recovery framework for identifying the governing equations of forced nonlinear dynamical systems under low sampling frequency constraints. The method integrates an information-theoretic analysis based on the Cramér–Rao lower bound and introduces a constrained equation search implemented via continuous-time latent variable networks (LTC-NN, CT-RNN, NODE). MRIDHA enforces stability and time-constant constraints during model identification, achieving improved sample efficiency and robustness near the Nyquist limit. The paper provides extensive theoretical discussion and large-scale experiments on both simulated and real-world datasets。

**Strengths:**

1. The paper evaluates the method on a wide range of benchmarks with systematic studies on sampling frequency, noise level, and input uncertainty. In the evaluation parts, the paper reports RMSE, MARD, and Hamming distance, and includes ablation studies for different network components (LTC-NN, CT-RNN, NODE).

2. The proposed method empirically improves model recovery near Nyquist-rate sampling, a regime where many existing methods fail.

**Weaknesses:**

1. The paper attempts to cover too many aspects (information theory, inverse ODE solution, latent dynamics modeling, etc.) within a single manuscript. The main theme of sample-efficient model recovery gets diluted among numerous details.

2. Although the paper includes real-world examples such as biomedical and EEG data, the experiments mainly report aggregate error metrics (RMSE, MARD, etc.) without demonstrating whether the recovered models capture domain-specific, physically or physiologically meaningful behaviors. For example, in the EEG setting, an important question is whether the identified model can accurately predict key spectral or temporal features, such as the timing of gamma-band peaks or cross-frequency phase–amplitude coupling patterns, which are critical problems in related tasks such as deep brain stimulation.
Simply showing that MRIDHA achieves lower RMSE than competing methods does not necessarily imply that the learned dynamics are valid or useful for neuroscientific interpretation. In fact, based on the reported metrics, it appears that none of the tested methods—including MRIDHA—achieve the level of predictive fidelity required for practical EEG or neural modeling tasks.

3. The baselines are primarily SINDy-MPC and PINN+SR. However, there exist recent system inference/reconstruction methods, such as [R1],[R2], which should be discussed or compared to position this work relative to the state of the art.

__References__

[R1] Course, K., & Nair, P. B. (2023). State estimation of a physical system with unknown governing equations. Nature, 622(7982), 261-267.

[R2] Oh, Y., Lim, D. Y., & Kim, S. (2024). Stable neural stochastic differential equations in analyzing irregular time series data. arXiv preprint arXiv:2402.14989.

**Questions:**

In the Supporting Argument, could the authors clarify the derivation leading to Equation (4)? Specifically, why does the upper bound estimation depend only on $\Theta_{est}$ (the estimated parameter) and not on the true parameter $\Theta$? A more transparent justification or intermediate steps would help verify the correctness of this bound.

---

### Official Review · Reviewer_VqpT · 2025-10-31

**Soundness:** 2
**Presentation:** 1
**Contribution:** 3
**Rating:** 2
**Confidence:** 4

**Summary:**

The authors propose a model discovery framework that infuses "DHArma" (which is never defined) in order to search an equation space to select models that respect physical properties such as stability and time constants.  The architecture is a bit confusing and is motivated by their rigorous information theoretic analysis.  This breaks down the model recovery into two distinct pieces:   a data-fit component (linear in sampling frequency) and a model-estimation component (nonlinear in frequency).

**Strengths:**

There are some good ideas in this paper, but the paper is so scattered in its narrative, starting with bounds and then proposing an algorithm based upon Hasani (2024).  I do think there are good ideas here, but his was the most difficult of all my papers to read given how scattered and disorganized it was.

**Weaknesses:**

The paper was quite incoherent, mixing the theory, computation and motivation in a way that was quite hard to track.  I would recommend the authors work to sort this out into a more coherent presentation.  Additionally, from the deployment perspective, it is not clear how much beyond Hasani 2024 the paper represents in terms of findings.   The example system and study also were quite scattered and lacked coherence in what was to be studied and compared across models. Certainly sampling, noise and model difficulty should be all considered.  Again, I think there are some interesting things in the paper, but it is just presented is such a haphazard way.

**Questions:**

The theory is fine in the paper, but is that what you want the reader to concentrate on?  Or the results from the method?
The comparisons are to SINDY-MPC and PINN-SR for instance, are these really the leading methods to be comparing too?

---

### Official Review · Reviewer_RSBs · 2025-11-02

**Soundness:** 3
**Presentation:** 1
**Contribution:** 2
**Rating:** 2
**Confidence:** 3

**Summary:**

In this paper, the authors present MRIDHA, which is a model discovery (recovery) method that is able to perform nonlinear system identification with fewer sample sizes. This is mostly performed via forward solving the discovered ODE with theoretical results on Carmer-Rao type of information lower bound. The authors further demonstrate several experiments including real-world and synthetic data to demonstrate the effectiveness of their method.

**Strengths:**

This paper is practically useful.

1. High temporal sampling rate can be difficult for some applications. It is important to know the theoretical limit, and improve sample complexity.

2. The usage of ODE integration is important and it is a valid method to improve sample complexity. The experimental result nicely shows that the algorithm is nicely designed.

**Weaknesses:**

The weaknesses of this paper are also very strong, and the reviewer feels regretful that this idea is being presented in this way. The presentation is definitely very suboptimal, which made

(i) The readers are unclear about the algorithm and why the algorithm would work.

(ii) The authors might also get lost without having these results clearly sorted out.

(iii) The experiments are also not nicely arranged so it looks very messy. It makes it hard for readers to tell what the conclusion is.

**Questions:**

1. The reviewer is not sure about the presentation of the paper writing in general. For example, in Table 1, most of the texts are in bold. It may confuse the readers and make people unsure what to focus on. In Table 7, it seems like the text RMSE, MARD, HD. It would be better to use \text{RMSE} if it is in a math environment.

2. The same problem might happen for the presentation of math equations, which can be important as a theory heavy paper. For example, in Page 3, right above Eqn. (2), t_1...t_N, t_i-t_{i-1}=1/f_s \forall i in \{1...N\} seem to be really dense to the reviewer.

3. In Eqn. (2), when we have an expectation - what is random here? I assume the X(t_i) is random but the fisher information, randomness, expectations are all not making any sense if the random variables are not defined here. The reviewer suggests the authors to clearly re-check Cramer-Rao bounds and rewrite this section with a clearer understanding in mind. Whatever behind can be meaningless without setting the notations correctly (for example what is sigma here)?

4. The reviewer suggests to use $\Theta_{\text{est}}$ for example to make the notation better.

5. Only a question: would authors believe that the ODE solver is really important here to improve the sample complexity? If so, more paragraphs should be centered around here, and show the audience that this is a critical point - and what benefits could we have from here (having an extensive study).

6. Experimentally, would the authors come up with a plan to better modularize the experiments? It would be better if things can be compact within one or two figures, or just within one table.

7. This paper contains an excellent and creative idea with strong potential impact. It is truly a pity that such a promising contribution is presented in a confusing manner, which prevented the reviewer from giving a higher rating. The reviewer strongly encourages the authors to carefully revise the manuscript, improving the clarity, organization, and presentation, which will greatly elevate this work.

---

### Official Review · Reviewer_FWF4 · 2025-11-08

**Soundness:** 2
**Presentation:** 3
**Contribution:** 2
**Rating:** 4
**Confidence:** 3

**Summary:**

This paper tackles a core challenge in modeling dynamical systems: how to recover the system when data is collected at or near the Nyquist sampling limit (and even lower) rather than densely oversampled. The work is a blend of theory and engineering: it sets an information-theoretic limit on what’s achievable with limited data and then designs a neural architecture that explicitly respects those limits. On the theory side, the authors analyze the Cramér–Rao lower bound for model recovery and show that reconstruction error fundamentally splits into two terms: data fitting and model estimation, and clarify which term is the main source of issue in different regimes. On practical side, They introduce MRIDHA, which builds physical consistency directly into the search space for candidate equations, more specifically, instead of unconstrained regression, MRIDHA embeds continuous-time latent variable (CTLV) nodes that enforce stability with input-dependent time constraints. They also demonstrate the effects on multiple simulated/real-world systems.

**Strengths:**

The paper makes an original connection between information theory and model recovery (MR), offering a clear theoretical framing via the Cramér–Rao lower bound and a practical method (MRIDHA) that enforces physical consistency for sparse data regimes. The architecture is thoughtfully engineered and clearly outperforms prior methods, demonstrated by its breath of experiments, ranging from chaotic systems to biomedical and high-dimensional control.

**Weaknesses:**

Despite the strong comparative results, I failed to find a quantitative check that MRIDHA approaches the predicted Cramér–Rao limit which seems to be the central point of the paper. (Or if it's not approaching that limit, what's the gap here?) Adding some empirical results on this point or sharing more insights on the gap would greatly strengthen the soundness of the work.

Another point I want to touch on is that, while the paper is technically rigorous, the underlying research question, i.e. recovering deterministic systems from limited samples may have diminishing practical importance in the current trajectory of data-driven modeling. In most real-world applications, especially in control, biomedicine, and robotics, approximate predictive fidelity or stability guarantees matter more than explicit symbolic recovery of dynamics. From experience in model discovery, once one leaves laboratory-scale systems, the true governing equations are often neither identifiable nor stationary due to unmodeled inputs, sensor drift, and environmental variability. Thus, the pursuit of sample-efficient exact recovery may be intellectually elegant but of limited operational relevance. A more consequential direction might be to extend the theoretical framework toward uncertainty quantification, for example, connecting the Cramér–Rao bounds to Bayesian confidence in recovered dynamics, or to explore sensor placement optimization under the same information-theoretic lens, asking how to actively allocate sampling budget for maximum information gain. Another valuable step could be to test MRIDHA in control-in-the-loop or adaptive filtering scenarios, where its structural priors might translate into measurable improvements in safety or robustness rather than only lower reconstruction error.

**Questions:**

It's unclear to me whether MRIDHA’s improvements truly stem from the imposed physical priors (like stability and input-dependent time constants) or from the added representational power and implicit regularization of the LTC-NN architecture? A natural question is: if SINDy-MPC or PINN-SR were augmented with similar ODE-based losses or CTLV-style temporal dynamics, would they close much of the performance gap?

---

### Note · Authors · 2025-11-12

I have read and agree with the venue's withdrawal policy on behalf of myself and my co-authors.